# Voluntary Carbon Disclosure (VCD) Strategy under the Korean ETS: With the Interaction among Carbon Performance, Foreign Sales Ratio and Media Visibility

**DOI:** 10.3390/ijerph191811268

**Published:** 2022-09-07

**Authors:** Sun Ae Kim, Jong Dae Kim

**Affiliations:** 1Sustainable Management Graduate Program, Inha University, Incheon 22212, Korea; 2Green Finance Graduate Program, Inha University, Incheon 22212, Korea

**Keywords:** carbon disclosure, carbon reduction performance, foreign sales, media visibility, sustainable report, CEI, ETS

## Abstract

While there has been a sufficient amount of research and empirical evidence on the factors that influence a company’s decisions to voluntarily disclose carbon information, little research has been done on the carbon disclosure practices of ETS-affected companies, in Asian countries, in particular. Considering this, it is essential to shed light on more diverse linkages between carbon performance and voluntary carbon disclosure (VCD) under ETSs taking into account the specific context of each individual country. Drawing on the Korean ETS-affected companies with a contents analysis of their sustainable reports from 2015 to 2019, the present research seeks to address the existing knowledge gaps in the current literature on carbon disclosure. In doing so, hierarchical ordinary least square (OLS) regression analysis is used to infer causality and assess the findings. The findings empirically prove a positive relationship between carbon performance and VCD, which means that the affected companies under the Korean ETS are likely to disclose more when they have favorable carbon reduction performance. Furthermore, this link tends to be amplified for companies with a high percentage of foreign sales, while the role of media visibility interacts differently with carbon performance in influencing VCD.

## 1. Introduction

Starting in the mid-2000s, companies worldwide began to provide information on their greenhouse gas (GHG) emissions, as well as their goals and strategies to reduce emissions [1]. Such data reporting has become known as voluntary carbon disclosure (VCD). Since then, VCD has been growing in importance, due to the increased demand for carbon reduction performance information along with the prospect that the information would improve financial performance. In this sense, the emergence of investors’ interests in climate-related financial risks calls for a specific type of carbon data pertaining to those risks to properly support rational investment decisions [2]. For investors, reliable climate data is crucial [3,4]. In response to these challenges, companies use various communication channels to do so, mainly through a sustainable report and the Carbon Disclosure Project (CDP).

Given the fact that an emission trading scheme (ETS) requires in nature measurements to be taken on financial accounting and the reporting of carbon emissions [5], it would be fair to say that companies affected by the Korean ETS adopted nationwide since 2015 would be expected to improve their level of disclosing carbon information in terms of both quantity and quality. Importantly, previous empirical research findings are divided and inconsistent. While some demonstrated a positive relation between an ETS and VCD [6,7], others showed opposite results [5,8,9]. However, it is also noteworthy that most prior studies on the disclosure associated with ETSs have typically focused on Western contexts or global levels, and very few have addressed VCD patterns influenced by carbon pricing mechanisms, particularly in Asian countries. Following the Paris Agreement in 2015, both developed countries and developing countries have been adopting carbon pricing initiatives, such as an ETS and carbon tax. In this sense, it is a must to shed light on more diverse linkages between carbon performance and VCD under ETSs by reflecting on the specific context of each individual country.

Motivated by the existing knowledge gaps, this study specifically addresses the following research question; How can the relationship between carbon performance and voluntary carbon disclosure be shaped under the Korean ETS? Additionally, are there any specific factors unique to Korea that play critical roles in determining the relationship?

Along with these research questions, the main purpose of this research is to identify the relationship between carbon performance and VCD by analyzing the disclosure practices of affected South Korean companies under the Korean ETS. This is achieved mainly through the lens of a strategic disclosure perspective, by going beyond the two competing theories: the legitimacy theory versus the voluntary disclosure theory. A further goal of this study is to articulate the relationship between carbon performance and VCD that could be moderated by external (strategic) factors which, include foreign sales ratios and media visibility, both of which are particular to the South Korean business landscape. 

Thus far, the legitimacy perspective has been dominant in this vein of the literature. This rationale claims that poorer performers are inclined to disclose more as a method to find excuses for their poor performance and, in doing so, securing legitimization from society [10,11]. However, over the past few years, empirical research on the relationship between carbon performance and VCD has shifted from a reactive approach to a more proactive and strategically based one, which involves disclosing better quality and more reliable carbon information. Among these rationales, ref. [4] determined that a strong environmental performance was significantly associated with environmental disclosures and economic performance, according to the voluntary disclosure theory. In this current study, institutional theory is explored as a complementary theoretical background for providing the foundation for a strategic disclosure approach enhanced by a national ETS, an institutional setting. The institutional view suggests that the legitimation intent of corporates using VCD is compromised in strongly carbon-regulated institutional contexts [9].

Based on a comprehensive and in-depth analysis of sustainable reports of affected companies in the Korean ETS, this study makes important contributions in two primary ways. The first is that, through a more disaggregated analysis of the nexus between an ETS and VCD, the current study presents diverse aspects and a phased procedure of voluntary disclosure practice under the institutional environment (ETS). By identifying the roles of two decisive factors which shape a Korean company’s response, foreign sales ratio and media visibility, this study challenges the idea that the quality of VCD is unconditionally improved under an ETS, and provides information on the distinct outcome in the Korean business context. The second contribution of this work is that the content analysis of sustainable reports allows the present research to identify the practices of Korean companies’ disclosures by sector and scoring index, and to simultaneously suggest practical policies for inducing them to better conduct moving forward.

The remainder of this paper proceeds as follows; the Section 2 provides an overview of prior studies on the issue and explains the hypotheses development. Next, Section 3 describes the methodology, sample and data collection. The findings are presented in Section 4. Finally, the Section 5 offers concluding remarks and reflects on the wider implications of the study’s findings in real-world settings.

## 2. Literature Review and Hypotheses

### 2.1. Carbon Performance and Voluntary Carbon Disclosure (VCD)

Some previous studies found that poor environmental performers tend to disclose more environmental information [12,13], whereas other studies demonstrated that better performers disclose more [4,14], and some found no significant association between the two constructs that relate performance to disclosure [11]. Therefore, since both good and bad environmental performers feel the need to provide information through disclosure, it may be reasonably posited that environmental performance is not a significant driver for environmental disclosure. The business management literature on this issue underscores external factors, such as supply chain pressures, shareholder actions and regulatory threats, including the threat of direct economic consequences, as drivers of proactive corporate disclosure of both climate change strategies and GHG emissions [15]. However, previous research already found that company size, age of assets, governance and listing status are far more significantly related to environmental disclosure. These company-specific determinants reflect that, essentially, companies tend to perceive environmental disclosure as an instrument of their legitimacy that serves to manage their reputations and shows compliance with social norms in order to gain said legitimacy. In this way, companies can ensure continued access to resources and secure their survival and success [16]. So, companies may disclose information to manipulate or educate stakeholders to obtain their support or approval, as it is often easier to manage a company’s image than to make tangible progress in environmental performance [17].

#### 2.1.1. Two Competing Theories

Essentially, voluntary disclosure is a response to the external social pressure which agrees with the principles of legitimacy theory. Nevertheless, traditionally previous research on the environmental performance–environmental disclosure relationship has yielded mixed results. The legitimacy theory predicts a negative association between environmental performance and disclosure, while the voluntary disclosure theory predicts that this relationship will be positive. Most studies have regarded these theories as offering competing perspectives [18,19]. Following the legitimacy perspective, voluntary disclosure appears to aim first to improve corporate image among stakeholders [10,20]. External pressures may encourage businesses to adopt a symbolic and superficial approach that is not motivated by genuine concern for improving performance and transparency [13,21]. Put simply, companies use disclosure to highlight actions taken to bring its performance up to societal expectations, or to justify any shortfall [16]. As climate change has garnered even more attention at the global level in recent years, companies with high carbon emissions are expected to face growing demand to demonstrate planned reduction performance. In this sense, the legitimacy view would suggest that greater public and regulatory scrutiny and demands will be met by more disclosure. It further implies that poor environmental performers will have a greater need to justify themselves through disclosure.

Regardless of the true motivations of corporate disclosure, VCD is indeed a vital part of corporate carbon management, and the reliable reporting of information is becoming an important aspect. Accordingly, empirical studies have aimed to assess the gap between disclosure and performance and to analyze whether disclosure reflects actual corporate performance [22]. Taking this claim one step further, and viewing it through the lens of voluntary disclosure theory, creates a way to see a proactive rationale that shows how good carbon performers disclose more carbon information. This line of reasoning has been increasingly adopted for explaining recent corporate disclosure patterns. Good environmental performers have a number of reasons to inform stakeholders of their environmental activities [23]. Companies want to call greater attention to improvements made to environmental programs and to capture the implicit endorsement of key stakeholders in the process. Responsiveness to CSR through activities that reduce environmental impact, for instance, can lead to a competitive advantage if a company is able to focus stakeholder attention on the resource in question [24]. In other words, voluntary disclosure theory is a strategy-based approach that predicts a positive association between environmental performance and the level of voluntary environmental disclosure. Companies adopt a posture of full disclosure because they fundamentally believe that their strengths outweigh their weaknesses, and they are committed to environmental disclosure as a matter of value. Being highly perceived as legitimate, an environmentally conscious company typically improves access to needed resources [25]. In fact, the legitimacy-based and voluntary disclosure approaches are fundamentally similar, in that they both seek to favorably manage disclosure, but do so from different operational conditions–one focused on maintaining legitimacy, and the other on strategic position [18].

#### 2.1.2. Nexus between ETS and VCD

The Paris Agreement, which was adopted in 2015 and came into force in 2016, has urged not only companies but also countries to take a more proactive stance toward a low-carbon economy. There are now 61 carbon pricing initiatives in place or scheduled for implementation, consisting of 31 ETSs and 30 carbon taxes, covering 12 gigatons of carbon dioxide equivalent (GtCO2e), or about 22% of global GHG emissions as of 2020 [26]. 

South Korea, the world’s seventh-largest emitter of GHG as of 2018, announced its own voluntary medium-term mitigation goal to reduce GHG emissions by 40% of its 2018 level by 2030. As one of the key measures to achieve this national goal, South Korea launched an ETS in 2015, which was the second nationwide “cap-and-trade” scheme introduced into operation since that introduced in Kazakhstan, Asia [27]. The Korean ETS is currently running phase III for a five-year period, having started in 2021. Phase III features a tighter cap and changes to allocation. It introduced changes to allocation and increased auction shares. Since January 2019, entities covered by the KETS not considered vulnerable to carbon leakage are obligated to acquire 3% of their allowances for compliance at auctions. This share is predicted to increase to over 10% during the period of 2021–2025, with further increases envisaged after Phase III. The role of grandfathering is set to decline over Phase III. The Korean government indicated that free allocation, based on sector-specific benchmarks, is expected to rise to at least 70% over the period of 2021–2025. This was intended to continue a shift toward benchmarking that was expected to reach 50% by 2020. The third phase is also likely to see the introduction of new derivatives products, and financial institutions would be able to participate in the emissions market, which could boost liquidity. More information of the Korean ETS is seen in Figure 1. 

The ETS’s intrinsic traits are bound to generate a higher quality of disclosure. On the one hand, the institution of an ETS in nature requires the development of corporate accounting and technical expertise to manage and to report on CO_2_ emissions [29]. This serves two important purposes: (1) The monitoring of actual emissions and the use and trading of emission allowance, so that regulators can ascertain whether companies are emitting within their stipulated caps and whether the purpose of the ETS is being served, and (2) to inform investors in the emitting companies. The presence, use and purchase/sale of emission allowances are economic activities that affect the financial performance and net worth of companies. Carbon reduction, or the lack thereof, can be a source of financial risk, and it therefore needs to be monitored by company shareholders and debtors [5]. On the other hand, it is also noted that an ETS can be considered as a soft law from an environmental disclosures perspective, since this mechanism does not directly regulate environmental disclosures practices; rather, it gives freedom to companies in practices regarding environmental disclosure and provides strategies for dealing with uncertainty [7]. From a practical point of view, once the government applies such mechanisms with a view to curb national emissions, companies also react proactively through reporting voluntarily. [30] found that when regulations bind companies to disclose carbon emissions, and this information subsequently becomes public, companies provide more disclosures, although these disclosures are not mandatory. Ultimately, an ETS paves the way for improving disclosure through strategic company approaches, as companies are inclined to shift from engaging in passive approaches to pursuing proactive environmental management that will include a higher quality of disclosure practices. Under such an ETS, nothing other than a company’s carbon reduction performance is highlighted and, additionally, with an ETS’s basic requirements for measuring and reporting their performance, companies come under pressure to find new strategies to deal with this environment.

#### 2.1.3. Complementary Theory

Institutional theory has also been cited in plenty of literature [31,32] to explain the formal or informal pressures of institutional arrangements on corporate disclosure practices. Institutional theory is based on the concept that organizations are influenced by their ‘intangible’ institutional environment and are required to conform to the collective norms and beliefs of that environment. Organizations must shape their institutional image to gain legitimacy and, consequently, gain access to resources. Therefore, the survival of organizations depends on their adherence to institutionally defined rules and norms [33,34]. One main component of institutional theory is the concept of institutional isomorphism, or the extent to which organizations have to conform to institutional norms and become similar to other organizations to gain legitimacy [33]. With greater carbon regulatory pressures at the national and regional levels, the institutional claim has become increasingly highlighted. Thus, through the lens of the institutional perspective, rather than simply adhering to a socially constructed environment, it becomes much more warranted that actors may purposefully exploit institutional demands to advance their own organizational goals [35]. Such strategic approaches point to more active, and even proactive, responses to institutional pressures [36], meaning that it would result in a company’s positive financial performance in the end. Taking national carbon regulations, such as an ETS, as an institutional setting, it describes that institutional pressures (coercive, mimetic and normative) strongly motivate companies within a field to adopt similar disclosure practices and, therefore, improves the overall quality of disclosure (or raises the overall limit line). Combining with all these stated factors, [9] empirically argues that, ultimately, the usage of disclosure as merely a legitimating tool is weakened in the ETS. Thus, the gap between the level of disclosure and actual environmental performance is reduced accordingly. This means that under the ETS, the relationship between carbon performance and disclosure is likely to become a positive link.

As such, the two competing rationales and the one complementary claim provide conflicting predictions on how carbon performance may affect discretionary voluntary carbon disclosure under an ETS.

Hence, in accordance with the main research question, the following null hypothesis is proposed: 

**Hypothesis** **1**$\;(H_{1})$**.** *Carbon performance will not be associated with voluntary carbon disclosure under the Korean ETS.*

### 2.2. Carbon Disclosure Strategy: How Internal Factors Interact with External Factors

When all of these stated claims are combined, it can be reasoned that an ETS and its attributes allow the ETS to lay the foundation of a company’s proactive carbon management approach by changing passive and reactive approaches into proactive and strategic-based ones and, consequently, raising the overall limit line (or increasing the overall level of the disclosure quality) through institutional pressures (coercive, mimetic and normative). Meanwhile, the mixed findings on the nature of the relationship between environmental performance and disclosure may suggest a further possibility; the relationship may not be straightforward, but may be significantly more complex instead. In other words, there could be some other factors that govern this relationship through complex moderating or mediating roles. In order to comprehend these complexities, the present study attempts to understand the role of other relevant variables as they might affect environmental performance and disclosure in the South Korean business landscape. Ultimately, it is also worth noting that the interaction effects of these variables could play a key role in shaping the recent carbon disclosure strategies of Korean companies.

A number of studies examined environmental disclosure motives and determinants, but less is known about how internal factors combine with external factors to promote said disclosure [23]. Therefore, the present research adopts carbon performance as an internal factor and foreign sales ratio and media visibility as external factors for attempting to combine these two different streams. Thus far, the characteristics of Korean companies in relation to VCD can be viewed in two main ways. First, most large companies tend to disclose more non-financial information in their sustainable reports. Second, for these large companies, there are two additional factors common to both low- and high-carbon performers: foreign sales ratio and media visibility. It is, therefore, warranted that these two factors are considered as moderating variables in this study.

#### 2.2.1. Foreign Sales Ratio, Carbon Performance and Disclosure

Ref. [37] argued that the level of internationalization of a company can lead to increased CSR and, in the case of this study, it can lead to increased VCD efforts. They denote that “…as businesses trade in foreign countries, they see the need to establish their reputations as good citizens in the eyes of new host populations and consequently will engage in CSR as part of this process”. Similarly, [38] as well as [39] posited that a company’s presence in foreign markets means that it is bound to disclose more comprehensive information in line with the reporting rules of the foreign business system. [40] offered empirical support that international presence can be a strong determinant for non-financial disclosure.

Exports account for 40% of Korea’s gross domestic product (GDP), which is 10 % higher than the global average of 30%. The trade dependence, which is the value obtained by dividing total imports and exports by GDP, was 63.5% as of 2019; the second highest among the G20 countries after Germany (www.kostat.go.kr) (accessed on 10 November 2021). Given this, it is reasonable to assume that the interaction effect of the foreign sales ratio on the relationship between carbon performance and disclosure, which is the main effect of this research, is likely to occur. Thus, companies developing a large part of their sales on the global stage will be subject to higher scrutiny from their trading-partner countries regarding their GHG emissions. This becomes especially true if the trading partners are from industrialized countries [41].

As companies that fail to respond to these demands for information can see their legitimacy threatened, along with their access to needed resources [4,20], one can expect that those companies with a higher percentage of international sales will have a higher probability of providing environmental information regarding GHG emissions through voluntary disclosure. This positive correlation was established by [42] in the case of US companies included in the S&P 500 Index. Accordingly, the current study highlights the indispensable role of proactive disclosure practices, among other factors, and evaluates how the foreign sales ratio serves as a moderator between carbon performance and subsequent changes in carbon disclosure.

Therefore, these rationales lead to the following hypotheses:

**Hypothesis** **2**$\left(H_{2}\right)$**.** *Foreign sales ratio will be positively associated with voluntary carbon disclosure.*

**Hypothesis** **3**$\left(H_{3}\right)$**.** *Foreign sales will moderate the relationship between carbon performance and voluntary carbon disclosure such that companies will be more likely to provide disclosure (improve the quality of disclosure) when foreign sales are higher.*

#### 2.2.2. Media Visibility, Carbon Performance, and Disclosure

Visibility affects the level of outside pressure a company experiences because stakeholders take greater interest in companies that are visible [25]. As press exposure increases, so does public scrutiny of the company in question, and this leads to corporate adjustments designed to placate stakeholders. Stakeholder pressure for action on environmental issues depends on public attitudes/sentiment that are/is, in turn, affected by media coverage. The stakeholder–media coverage dynamic has important implications for voluntary environmental disclosure; the increased media coverage of climate science and environmental policy heightens the role of NGOs and signals a shift in public opinion. Consequently, it is conceivable that the pattern of media coverage simultaneously reflects and shapes corporate disclosure strategies. In other words, disclosure strategies of companies are likely to be shaped by the institutional environment, which is in turn affected by media coverage [23].

Following this line of reasoning, companies with neither low nor high levels of environmental performance are less visible and, consequently, better positioned strategically to withhold disclosure than either their lower or higher performing peers [18,23]. Ref. [43] went so far as to suggest that some environmentally legitimate companies are reluctant to tout their environmentally responsible activities and stop disclosure after they achieve legitimacy to avoid additional visibility and the added scrutiny. Ref. [44] also stated that companies choose to basically aim to meet, rather than exceed, the expectations of institutional stakeholders and social actors.

By identifying the extent to which voluntary disclosure is associated with visibility, ultimately, the study helps increase the understanding of how external pressures (from the media and public attention) encourage corporate compliance with voluntary initiatives or shape such voluntary disclosure strategy. To this end, specifically, this study proposes that visibility has a direct association with the level of voluntary carbon disclosure and interacts with carbon performance to strategically influence the relationship between carbon performance and voluntary disclosure. 

Hence, the hypotheses are developed as follows:

**Hypothesis** **4**$\;(H_{4})$**.** *Media visibility will be positively associated with voluntary carbon disclosure.*

**Hypothesis** **5**$\;(H_{5})$**.** *Media visibility will moderate the relationship between carbon performance and voluntary carbon disclosure such that companies will be more likely to provide disclosure (improve the quality of disclosure) when media visibility is higher.*

See Table 1 for a summary of all hypotheses mentioned in the previous sections.

## 3. Research Design and Methodology

### 3.1. Sample Selection

The Korean companies analyzed in this study were (a) part of the Korean emission trading scheme (ETS) and the Korean target management system (TMS) from 2015 to 2019, (b) listed on the Korean Exchange (KRX), and (c) had published a sustainable report during the same period. The sustainable reports of those affected by the Korean ETS were used for the purpose of this study, since they represent corporate communications and are some of the most important means of self-presentation. 

In total, the sample comprised of 421 yearly GHG emission data sets and 397 reports from 94 companies between 2015 and 2019. The companies belonged to nine industry sectors; materials, industrials, energy, consumer discretionary, consumer staples, IT, telecommunications, healthcare and finance, in accordance with the global industry classification standard (GICS). 343 observation sets were ultimately analyzed. To attain the study’s objectives, hierarchical ordinary least square (OLS) regression analysis was used to infer causality and assess the relationship between carbon performance and VCD and how the moderating effects on this relationship would result. All variables, except dummy variables, were winsorized at 1% and 99%.

### 3.2. Data Measures

#### 3.2.1. Voluntary Carbon Disclosure

This study derived the dependent variable, the quality of VCD, from sustainable reports of those South Korean companies affected by the Korean ETS (including those affected by the target management system). Reports for the financial years from 2015 to 2019 were selected. The reports in question were solely standalone reports listed on company websites. Mainly, the sections on climate change from the sustainable reports were analyzed based on the content analysis criteria, which were derived from the carbon disclosure project (CDP) and further modified [45]. The criteria and individual items are listed in Table 2.

Each criterion was given a score of 0, 1 or 2 depending on whether the criterion was deemed to be ‘not disclosed’, ‘partially disclosed’ or ‘fully disclosed’, respectively. Scoring was carried out according to a set of predefined rules for each criterion. The scores were then aggregated to achieve an overall disclosure quality score. As per the index, the maximum score a company could obtain was 36, given the 18 items. Finally, the range of the raw score varied from 0 to 36. Similar to the studies of [7,46], this study used a disclosure score by arriving at a relative value that was the raw score of an individual company divided by the median raw score of each industry a company belongs to, as follows Equation (1):(1)$$\mathrm{VCD}\_\mathrm{qual}=\frac{Raw\;Score}{Median\;Score}$$

#### 3.2.2. Carbon Performance

In order to measure the independent variable, carbon (reduction) performance, the Korean Greenhouse Gas Inventory and Research Center databases were used as sources for companies affected by the Korean ETS. The Korean ETS regulates direct/Scope 1 and indirect/Scope 2 ${\mathrm{CO}}_{2}$ emissions, in contrast with the EU ETS which controls only for Scope 1. This research measured carbon performance based on carbon emission intensity (CEI). A higher value for carbon emission intensity suggests that a company uses its resources, particularly energy, inefficiently, therefore making it a poor carbon performer [3]. CEI was calculated as the ratio of total GHG emissions of Scopes 1 and 2 to the total sales of a company, which reflects the efficiency of its production processes. The use of total sales as a scale (measure) is consistent with prior studies [14,47]. In addition, the natural logarithm transformation was employed due to the highly skewed distribution of emission-intensity variables [4,48]. Hence, the measurement describing carbon performance achieved by a company over time was as follows in Equation (2):(2)$${\mathrm{CP}}_{i,t}\;=\;\log\left(\frac{{\mathrm{EMI}}_{i,t}}{S_{i,t}}\right)$$
where ${\mathrm{CP}}_{i,t}$ denotes the carbon performance of company *i* in time *t*, ${\mathrm{EMI}}_{i,t}$ denotes its carbon emissions and $S_{i,t}$ denotes its sales.

#### 3.2.3. Foreign Sales Ratio

It is a well-accepted rationale that companies with higher foreign sales face a greater pressure for proactively engaging in reducing carbon emission and disclosing information. Therefore, foreign sales ratio is an indispensable factor to be considered when a Korean company designs their corporate strategy, given the Korea’s heavy dependence on exports. It was measured as the total foreign sales to the total sales of a company in a given fiscal year [42].

#### 3.2.4. Media Visibility

Drawing on [49], this study established two primary assessments to determine visibility: relevance and prominence. It selected three South Korean daily newspapers, the JoongAng Daily, Dong-A and Mail Economics Daily News from 2015 to 2019, because they are leading political, business and social news outlets that cover the full spectrum of issues and events. After conducting text-mining with the use of the coding program “Python”, the current study allocated one point for each time the name of the company was mentioned within an article (relevance), and an additional point if the name appeared in the headline of an article (prominence). In this way, a story containing a company’s name in the title would receive a score of “2” for visibility, and this research totaled the scores for each company.

#### 3.2.5. Control Variables

Company size used a natural logarithm of total assets as a proxy which is in line with prior studies [6]. The company’s financial performance used a market-based variable which was return on equity (ROE) [18]. This study used the ratio of a company’s total debt to its total assets as a proxy for leverage [48] and, also, capital expenditure as a proxy for influencing strategic planning [50]. To control for industry differences, the selected companies belonged to nine sectors, all of which were further classified into three main industrial groups [5,51]: industrial, material and energy sectors were categorized into Group 1, Group 2 comprised consumer discretionary and consumer staples and Group 3 included IT, telecommunication and healthcare, with the financial industry as the base group. Therefore, three dummy variables were set in this study. If a company belonged to a certain group, it was assigned a value of 1, otherwise it was assigned 0 [18,52]. Four dummy variables indicating the five years of the sample; Year 2015, Year 2016, Year 2017 and Year 2018, with Year 2019 as the base year; were used to control the time-specific factor. As lager emitters have been under increasing pressure, they have become highly involved in environmental management, including disclosure [53]. Thus, this study controlled for a company’s total amount of GHG direct/indirect emission (Scopes 1 and 2) at the fiscal year. Those who submit a response to CDP questionnaires annually have proven to produce higher-quality sustainable reports [54]. If a company participates in CDP scoring on an annual basis, it was assigned a value of 1, and it was assigned a value of 0 otherwise.

Based on the main relationship between carbon performance and disclosure quality discussed previously, the study estimated the following model, Equation (3): (3)$${\mathrm{VCD}}_{i,t}=\beta_{i}+\beta_{1}\;CP_{i,t}+\beta_{2}\;F\_sales_{i,t}+\beta_{3}\;M\_visi_{i,t}+\;\;\beta_{4}\;(CP_{i,t}\times \;F\_sales_{i,t\;})\;+\;\beta_{5}\;(CP_{i,t}\times M\_visi_{i,t})\;+\;\beta_{6}\;SIZE_{i,t}+\beta_{7}\;Lev_{i,t}+\;\beta_{8}\;CapEx_{i,t}\;\;+\beta_{9}\;ROE_{i,t}\;+\;\;\beta_{10}\;EMI_{i,t}+\;\beta_{11}\;IND+\;\beta_{12}Y+\beta_{13}CDP\;+ɛ$$

## 4. Result and Analysis

### 4.1. Descriptive Analysis

Table 3 indicates that the average total emission intensity was 0.489, which means that, on average, the sample companies generated 0.489 tons (489 kg) of total carbon emissions per one million KRW of sales. Given the previous study’s finding that showed the mean value of 0.516 from 284 Fortune 500 companies from 2008 to 2012 [55], the general movement of reduction performance over the five-year study period, which is the initial stage of the Korean ETS, can be seen as fairly encouraging. On the other hand, the mean value of VCD over the five-year period was 14.974, indicating that, on average, companies consistently maintained their disclosure scores at around 15 (out of 36) per year. This trend suggests that regardless of company motivations, the Korean ETS-affected companies have maintained a moderate level of disclosure quality over the study period.

Furthermore, the current research explores VCD by companies in different industries (Table A1). Also, it presents the percentage of companies that provided the information satisfying each item (Table A2). Given the recent discussion of the task force on climate-related financial disclosure (TCFD) guidelines regarding the improvement of VCD, this percentage distribution of disclosure score can provide important implications and suggest which areas need to be further developed. The results for media visibility in Table 4 show that there was a lot of variability (high standard deviation) noted in the amount of media exposure among the sample companies with, perhaps unsurprisingly, the size of the companies being the biggest differentiator. It is also expected that companies close to the market and those that have consumer facing activities will be more open to media attention, compared with companies that are non-branded and involved in mainly business to business activities [56]. Correlation coefficients are pooled in Table 5.

### 4.2. Regression Analysis

The results of the hierarchical regression analysis with disclosure quality as the dependent variable for Hypotheses 1–5 are presented in Table 6. Since we used a panel data, we conducted a Hausman test detecting fixed or random effects in the regression. It turned out that the fixed effect model was more appropriate than a random model. Thus, we included both yearly and industry dummy variables.

Model 1 includes the main independent variable, CEILog and the control variables. The model confirms that a company’s carbon performance significantly influences its disclosure quality (F = 10.232, *p* < 0.001), as the yearly improvement in a company’s carbon performance is negatively associated with higher disclosure (β = −0.203, t = −3.249, *p* < 0.01). Thus, null Hypothesis 1 $\left(H_{1}\right)$ is rejected. As the actual carbon performance should be read as the inverse of the variable (CEILog), the smaller the measure is, the better the carbon performance of the company is. This provides strong support for the voluntary disclosure theory perspective, that shows how companies with a good carbon performance are likely to raise the disclosure score in order to promote their better output. In Model 1, most of the control variables, except leverage, CDP performance and year dummy variables are found to be significant predictors of disclosure quality, thereby showing that larger companies with a good financial performance emit more GHG emissions.

Model 2 and Model 3 in Table 6 include the main effects of carbon performance (CEILog), the main effects of moderating variables, foreign sales ratio (F_sales) and media visibility (M_visi), as well as interactions between them. Hypotheses 2, 3 and 4, 5 assumed that the relationship between carbon performance and disclosure quality would be moderated by foreign sales ratio and media visibility, respectively. To this end, the hierarchical regression firstly tested the main effects of foreign sales ratio $\left(H_{2}\right)$. Due to the fact that Hypotheses 2 $\left(H_{2}\right)$ and 4 $\left(H_{4}\right)$ posed positive relationships, respectively, between foreign sales ratio and media visibility and disclosure quality, the correlation for the foreign sales ratio and media visibility variables must be positive and statistically significant for Hypotheses 2 and 4 to be supported.

With the fulfilled premise (Table 5), Model 2 empirically shows that the main effect of carbon performance is significant, and that overall effect is still negative (β = −0.273, *p* < 0.001). The main effect of foreign sales ratio is also significant and positive (β = 0.229, *p* < 0.001), as was expected in the correlation; thus, Hypothesis 2 $\left(H_{2}\right)$ is accepted.

In Hypothesis 3 $\left(H_{3}\right)$, the positive moderation effect of the foreign sales ratio was assumed and, accordingly, the real effect of the interaction between CEILog and foreign sales ratio is found to be significant and negative (β = −0.129, *p* < 0.05). The result empirically implies that companies with a high foreign sales ratio are likely to generate high quality disclosures driven by their improved carbon performance, while they tend to disclose less carbon information when they perform poorly. Accordingly, Hypothesis 3 $\left(H_{3}\right)$ has been fully supported. This moderation effect result strengthens the main negative relationship between carbon performance and disclosure quality. Model 2 empirically proves that Hypotheses 2 and 3 are firmly accepted. 

Model 3 mainly tested the interaction effects of the second moderating variable ($H_{4}\;$ and $H_{5}$), media visibility. In doing so, it firstly shows that the main effect of carbon performance is significant (β = −0.171, *p* < 0.05), and still trends in a negative direction. The main effects of media visibility are also significant but positive (β = 0.245, *p* < 0.001), which is consistent with the correlation (Table 5). As such, these findings lay a firm foundation for testing Hypothesis 5 $\left(H_{5}\right)$, which depicted interactions between media visibility with carbon performance as well. Hypothesis 5 $\left(H_{5}\right)$, which stated that companies will be more likely to provide disclosure when media visibility is higher, is not entirely, but fairly, confirmed in Model 3. 

Its interaction effect is proven to be significant but negative (β = 0.156, *p* < 0.05), literally meaning that companies are less likely to provide when media visibility is higher, which is the opposite result to the assumed negative sign, in that the present research reverses the sign of the variable to facilitate the interpretation of carbon performance measured by CEI. In this sense, the result suggests that the (main) relationship between carbon performance and disclosure in Hypothesis 5 $\left(H_{5}\right)$ is expected to weaken in accordance with the moderation effect of media visibility. Put differently, based on Hypothesis 5 $\left(H_{5}\right)$, if companies with high media visibility perform favorably in carbon reduction, they would invariably produce a higher quality of disclosure to tout their better carbon performance and gain a competitive advantage. However, based on the aforementioned findings, companies with high media visibility become unwilling to promote their superior carbon reduction performance by stopping additional disclosure once they reach a certain level, at which point they can be unnoticeable.

This finding is consistent with the claim in the existing literature pertaining to the relationship between environmental disclosure and corporate visibility, as indicated in the hypotheses development section of this study. Typically, companies with higher media visibility (those either larger in size or heavier emitters) are bound to emit more, perform relatively better in GHG emission reduction, and disclose more fairly, heeding to external pressures. Hence, they don’t have any motivation to tout their carbon performance through additional disclosure, since it can draw extra scrutiny from the media, NGOs and authorities. It can also be interpreted that companies with high media visibility are likely to only meet the requirements of a level to the extent that they can secure anonymity in the ETS, to avoid additional external pressure.

## 5. Conclusions and Discussion

While there has been ample research with empirical evidence on the factors influencing companies’ decisions to voluntarily disclose carbon information, not much research has been undertaken on the carbon disclosure practices of ETS-affected companies, in particular in Asian countries. Furthermore, provision of or disclosure of financial and non-financial information pertaining to the impact of carbon emissions is arguably crucial for the operation of ETSs, and remains topical for standard setters and report users. Thus, the current study is one of the most recent to examine the relationship between carbon emission reductions performance and VCD for a sample of Korean ETS-affected companies by looking at the quality improvement driven by the ETS.

Amid the heated debate over the national legislation on environmental disclosure, which aims to come into effect in 2030 in South Korea, the importance of VCD in the Korean business environment has been more prevalent than ever. In the context of inconclusive association between carbon performance and VCD, the Korean ETS ultimately appeared to lay the foundation for a proactive corporate carbon management approach, by shifting from a passive and reactive approach to one that is strategically based. Specifically, such proactive strategic approaches were examined in detail in conjunction with the two external factors, foreign sales ratio and media visibility. To this end, two contrasting theoretical frameworks, legitimacy and voluntary disclosure perspectives, as well as a complementary basis, institutional theory, were explored.

As a result, under the ETS, with the weakened legitimating tool of disclosure, the current study empirically proves a positive relationship between carbon performance and VCD, meaning that the affected companies under the Korean ETS are likely to disclose more when they have a favorable carbon reduction performance. Furthermore, this link tends to be amplified for companies with a high percentage of foreign sales, while the role of media visibility interacts differently with carbon performance to influence VCD. 

The empirical analysis suggests that it is more inclined for disclosure to be adopted in a strategic way. Companies with high media exposure appear to be larger in size, emit more in general, and come under a higher level of external scrutiny. They have little to gain from more highlighting their carbon performance, and instead withhold disclosure because they are already meeting the basic level of performance expectations.

The current study makes both theoretical and empirical contributions to the existing body of research in its field. Firstly, by exploring strategic disclosure approaches to the relationship between carbon performance and VCD, this research contributes to shedding light on VCD strategy in accordance with corporate objectives and resources with a fresh approach. The current research, under the Korean ETS (coercive pressure), sought to combine an internal factor (carbon performance) and external factors (foreign sales ratio and media visibility) that has rarely been attempted in this field to clarify how companies respond strategically to uncertain and evolving policies. Secondly, this study recognizes the importance of the external factor, media visibility for seeking corporate VCD strategy. In contrast to [57], who did not find significant relationships between visibility and company behavior, the media visibility results in this study differ in that the relationship between carbon performance and VCD significantly varies depending on the level of media exposure under the ETS. More specifically, the level of company visibility determines the extent of corporate disclosure. This further enhances the strategic approach such that companies exceeding stakeholders’ expectations provide disclosure to highlight their favorable performance, but not so much disclosure as to invite additional media (or external) scrutiny [58].

Focusing on the carbon disclosure practices of ETS-affected companies via contents analysis of sustainable reports makes it possible to (1) identify which aspects of carbon-related disclosures are falling short of investor expectations and suggest areas of VCD wherein further development is essential; (2) explore the country differences in these carbon disclosure practices by comparing the VCD practice of recent ETS-affected companies from Asian countries with that of the older ETS-affected companies from European countries; and (3) reflect the industry/sector differences when preparing for the upcoming global pressures, for instance the EU’s carbon border adjustment mechanism (CBAM).

Despite examining a relatively longer time period (five years with annual data) than many other studies of carbon performance, the empirical analysis still covers a limited time span. If the study compared the pre-ETS and the post-ETS situation in South Korea, the explanation would have been more powerful. By illuminating the nature of the interface between carbon performance, ETS and VCD, it is hoped that this study will assist with the efforts of companies to provide reliable disclosure and the efforts of stakeholders to obtain the information.

## Figures and Tables

**Figure 1 ijerph-19-11268-f001:**
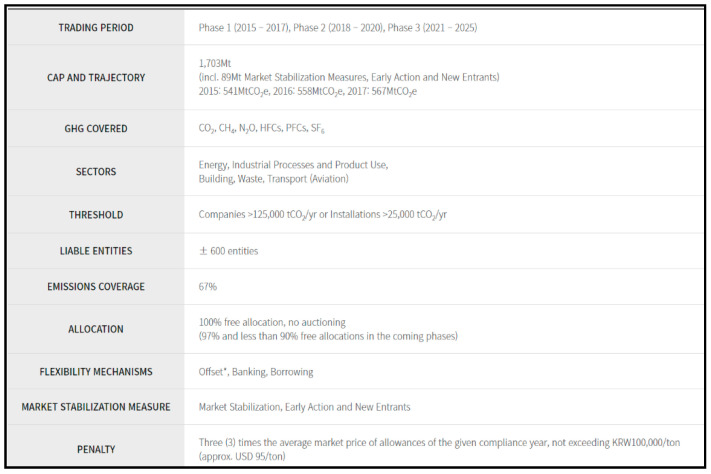
Korean ETS. Source: [28] ECOEYE, 2021.

**Table 1 ijerph-19-11268-t001:** Summary of Hypotheses.

Hypothesis	Description
Hypothesis 1 (*H*_1_)	Carbon performance will not be associated with voluntary carbon disclosure under the Korean ETS.
Hypothesis 2 (*H*_2_)	Foreign sales ratio will be positively associated with voluntary carbon disclosure.
Hypothesis 3 (*H*_3_)	Foreign sales will moderate the relationship between carbon performance and voluntary carbon disclosure such that companies will be more likely to provide disclosure (improve the quality of disclosure) when foreign sales is higher.
Hypothesis 4 (*H*_4_)	Media visibility will be positively associated with voluntary carbon disclosure.
Hypothesis 5 (*H*_5_)	Media visibility will moderate the relationship between carbon performance and voluntary carbon disclosure such that companies will be more likely to provide disclosure (improve the quality of disclosure) when media visibility is higher.

**Table 2 ijerph-19-11268-t002:** Contents Analysis Index for Scoring Sustainable Reports.

Categories	Items	
1.Climate Change risks and opportunities	CC1	assessment/description of the risks (regulatory, physical or general) relating to climate change and actions taken or to be taken to manage the risks
	CC2	assessment/description of current (and future) financial implications, business implications and opportunities of climate change
2. GHG emissions	GHG1	description of the methodology used to calculate GHG emissions (e.g., GHG protocol or ISO)
	GHG2	existence external verification of quantity of GHG emission—if so by whom and on what basis
	GHG3	total GHG emissions—metric tonnes CO-e emitted
	GHG4	disclosure of Scopes 1 and 2, or Scope 3 direct GHG emissions
	GHG5	disclosure of GHG emissions by source (e.g., coal, electricity, etc.)
	GHG6	disclosure of GHG emissions by facility or segment level
	GHG7	comparison of GHG emissions with previous years
3. Energy consumption	EC1	total energy consumed (e.g., tera-joules or peta-joules)
	EC2	quantification of energy used from renewable source
	EC3	disclosure by type, facility or segment
4. GHG reduction and cost	RC1	detail plans or strategies to reduce GHG emissions
	RC2	specification of GHG emissions reduction target level and target year
	RC3	emission reductions and associated costs or savings achieved to date as a result of the reduction plan
	RC4	costs of future emissions factored into capital expenditure planning
5. GHG emission accountability	ACC1	indication of which board committee (or other executive body) has overall responsibility for actions related to climate change
	ACC2	description of mechanism by which the board (or other executive body) reviews the company’s progress regarding climate change actions

Note. The index is described in greater detail by Choi et al. (2013).

**Table 3 ijerph-19-11268-t003:** Summary of Company Carbon profile.

Descriptive Stats	Carbon Emission Intensity	Disclosure score
Mean	0.48865	14.974
Std.dev.	1.52388	4.668
Min	0.002	4
Q1	0.02211	12
Median	0.06231	15
Q3	0.24877	17
Max	11.076	33
Sample Number	421	397

Note. 421 is the total sample number of CEI (the level of standardized carbon emissions using a company’s total sales for any given year) before being transformed into the log of Equation (1). 397 is the total number of sustainable reports without measuring the relative value of Equation (2).

**Table 4 ijerph-19-11268-t004:** Descriptive Statistic.

Variables	Mean	Std.dev.	Min	Q1	Median	Q3	Max
CEILog	−2.6511	1.86311	−6.11	−3.805	−2.775	−1.385	2.11
Disclosure quality	1.0154	0.30458	0.27	0.8421	1	1.1429	2.2
Foreign sales	0.3916	0.28009	0.0003	0.12	0.36	0.61	0.95
Media Visibility	155.842	254.35565	0	21	69.5	180.25	1450
Size	15.564	1.5573	12.08	14.58	15.49	16.5025	19.88
Leverage	0.491	0.20825	0.03253	0.35562	0.5063	0.6164	0.93928
CapEX	2.2365	3.3538	0.039	0.91	1.25	1.84	18.79
ROE	5.8247	30.6914	−1251.9	0.9775	5.95	10.1125	210.85
EMI	1,977,710	6,827,176	5980	54,471	157,533	802,266	59,573,482
N = 343							

**Table 5 ijerph-19-11268-t005:** Correlation Coefficient.

Variable	1	2	3	4	5	6	7	8	9	10	11	12	13	14	15	16	17
1. VCD_qual	1																
2. Size	0.386 ***	1															
3. Lev	−0.169 **	0.035	1														
4. CapEX	0.161 **	0.359 ***	0.077	1													
5. ROE	0.020	−0.036	−0.031	0.123 *	1												
6. EMI	0.373 ***	0.250 ***	−0.252 ***	−0.063	−0.008	1											
7. CDPper	0.211 ***	0.342 ***	−0.167 **	0.047	−0.012	0.109 *	1										
8. IND Group1	−0.021	−0.324 ***	−0.009	−0.194 ***	−0.203 ***	0.115 *	−0.027	1									
9. IND Group2	−0.095 *	−0.060	−0.096 *	−0.115 *	0.041	−0.1 *	0.052	−0.542 ***	1								
10. IND Group3	0.093 *	0.168 **	−0.156 **	−0.118 *	0.043	−0.009	0.010	−0.588 ***	−0.149 **	1							
11. Y2015	−0.025	0.037	0.050	0.011	−0.013	0.018	0.024	−0.020	0.015	0.015	1						
12. Y2016	0.039	0.034	0.011	0.026	−0.011	0.008	0.044	0.009	0.000	−0.025	−0.207 ***	1					
13. Y2017	0.001	−0.047	−0.038	−0.01	0.011	−0.010	−0.027	0.000	0.016	0.008	−0.231 ***	−0.244 ***	1				
14. Y2018	0.013	−0.033	−0.004	−0.032	0.009	−0.012	−0.034	−0.001	0.007	−0.001	−0.237 ***	−0.251 ***	−0.279 ***	1			
15. CEILog(CP)	0.016	−0.225 ***	−0.279 ***	−0.039	0.017	0.454 ***	0.033	0.294 ***	−0.218 ***	−0.009	0.026	0.019	0.016	−0.020	1		
16. F_sales	0.255 ***	0.113 *	−0.277 ***	−0.269 ***	−0.058	0.053	0.208 ***	0.083	−0.141	0.258 ***	−0.008	−0.017	−0.025	0.006	0.152 **	1	
17. M_visi	0.324 ***	0.503 ***	−0.124 *	−0.042 *	−0.001	0.116 *	0.208 ***	−0.398 ***	0.185 ***	0.375 ***	0.031	0.018	−0.021	−0.024	−0.119 *	0.197 ***	1

Note. Correlation is significant at * *p* < 0.05, ** *p* < 0.01, *** *p* < 0.001.

**Table 6 ijerph-19-11268-t006:** Regression Result.

Variables	*Dependent Variable: VCD_qual*		
	[Model 1]	[Model 2]	[Model 3]
	Coeff. t-Value	Coeff. t-Value	Coeff. t-Value
CEILog	−0.203 ** (−3.249)	−0.273 ***(−4.362)	−0.171 * (−2.273)
F_sales		0.229 *** (4.135)	0.218 *** (3.978)
CEILog * F_sales		−0.129 * (−2.419)	−0.114 * (−2.111)
M_Visi			0.245 *** (3.594)
CEILog * M_visi			0.156 * (2.281)
Size	0.304 *** (4.279)	0.216 ** (2.994)	0.128 (1.540)
Leverage	−0.005 (−0.089)	0.024 (0.416)	0.019 (0.346)
CapEx	0.341 *** (4.496)	0.377 *** (5.097)	0.395 *** (5.427)
ROE	0.114 * (2.161)	0.067 (1.275)	0.057 (1.094)
Emission Amount	0.365 *** (5.972)	0.420 *** (6.972)	0.321 *** (4.270)
CDPper	0.051 (0.957)	0.029 (0.567)	0.049 (0.962)
Industry:			
IND Group 1	0.818 *** (4.382)	0.549 ** (2.903)	0.468 * (2.487)
IND Group 2	0.483 ** (3.494)	0.343 * (2.502)	0.261 (1.867)
IND Group 3	0.631 *** (4.403)	0.420 ** (2.893)	0.299 * (2.021)
Year:			
Y2015	−0.011 (−0.188)	0.004 (0.065)	0.004 (0.081)
Y2016	0.048 (0.836)	0.066 (1.176)	0.059 (1.084)
Y2017	0.036 (0.612)	0.062 (1.081)	0.065 (1.158)
Y2018	0.052 (0.879)	0.066 (1.151)	0.067 (1.192)
Constant	−0.672 * (−2.262)	−0.381 (−1.298)	−0.024 (−0.074)
adju R²	0.317	0.367	0.394
F-static	10.232 ***	11.074 ***	10.980 ***
N	343	343	343

Note. Correlation is significant at * *p* < 0.05, ** *p* < 0.01, *** *p* < 0.001.

## Data Availability

Not applicable.

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
