# Peer review of "Voluntary Carbon Disclosure (VCD) Strategy under the Korean ETS: With the Interaction among Carbon Performance, Foreign Sales Ratio and Media Visibility"

_ijerph, 2022, doi:10.3390/ijerph191811268_

Round 1

Reviewer 1 Report

This paper can be a valuable contribution to the International Journal of Environmental Research and Public Health. The paper contributes with respect to the voluntary carbon disclosure strategy in the case of Korean ETS. The objectives and methodologies are in line with the target journal.

However, some improvements must be made to enhance the readability of the article and its implications for both the scientific community and policymakers. 

Below I list a set of indications that the author can consider: 

Title: the title should be improved and I suggest avoiding the use of ":" followed by "with". 

Introduction: the introduction is missing the presentation regarding the general structure that the paper follows (as the indication of its main paragraphs/contents).

Theoretical framework

2.2.1" Two competing theories". It would be best to specify already in the title the two theories that are competing and discussed.

In general, better link the paragraphs that seem a bit disconnected.

At the end of the theoretical framework it could be useful to include a figure/table summarizing the Hyphothesis.

The article is missing some key references that were published in the target journal and that have dealt with similar issues concerning also Korean cases. These articles (or similar ones) should be considered not only in the theoretical framework but as part of the discussion with respect to the results provided.  Kim E, Kim S, Lee J. Do Foreign Investors Affect Carbon Emission Disclosure? Evidence from South Korea. International Journal of Environmental Research and Public Health. 2021; 18(19):10097. https://doi.org/10.3390/ijerph181910097 Lee J-H, Cho J-H. Firm-Value Effects of Carbon Emissions and Carbon Disclosures—Evidence from Korea. International Journal of Environmental Research and Public Health. 2021; 18(22):12166. https://doi.org/10.3390/ijerph182212166 Tang Y, Zhu J, Ma W, Zhao M. A Study on the Impact of Institutional Pressure on Carbon Information Disclosure: The Mediating Effect of Enterprise Peer Influence. International Journal of Environmental Research and Public Health. 2022; 19(7):4174. https://doi.org/10.3390/ijerph19074174

Methodology

The division of single paragraphs for each variable (3.22 and so on) could be avoided by including one structured paragraph. 

Language:

ETS should be presented in an extended version the first time it is used. It is suggested to proofreading the article to improve its readability and clarity. 

Author Response

[Revision Summary Report]

 Response to First Reviewer 

(1)   : the title should be improved and I suggest avoiding the use of ":" followed by "with" 

     → Voluntary Carbon Disclosure (VCD) Strategy under the Korean ETS with the  Interaction among Carbon Performance, Foreign Sales Ratio and Media Visibility 

(2)  The introduction is missing the presentation regarding the general structure that the paper follows (as the indication of its main paragraphs/contents).

→ Page 4: the structure of the paper is included. 

(3)   Theoretical Framework 

Two competing theories". It would be best to specify already in the title the two theories that are competing and discussed.In general, better link the paragraphs that seem a bit disconnected.

 → Please refer to page 3 for the description of the competing theories.

At the end of the theoretical framework it could be useful to include a figure/table summarizing the Hyphotheses. 

   → Page 13: the table summarizing the Hypotheses is included. 

(4)   The article is missing some key references that were published in the target journal and that have dealt with similar issues concerning also Korean cases. These articles (or similar ones) should be considered not only in the theoretical framework but as part of the discussion with respect to the results provided.  

â‘    Kim E, Kim S, Lee J. Do Foreign Investors Affect Carbon Emission Disclosure? Evidence from South Korea. International Journal of Environmental Research and Public Health. 2021; 18(19):10097.   

â‘¡   Lee J-H, Cho J-H. Firm-Value Effects of Carbon Emissions and Carbon Disclosures—Evidence from Korea. International Journal of Environmental Research and Public Health. 2021; 18(22):12166.  

â‘¢   Tang Y, Zhu J, Ma W, Zhao M. A Study on the Impact of Institutional Pressure on Carbon Information Disclosure: The Mediating Effect of Enterprise Peer Influence. International Journal of Environmental Research and Public Health. 2022; 19(7):4174.  

   → I looked into the above three papers. Even though â‘ , â‘¡ deal with Korean companies’ carbon disclosure practices, they are different from our research approach. While our work has undertaken the association between carbon reduction performance and VCD by looking at a strategic disclosure approach enhanced by a national ETS, they primarily focused on factors influencing companies’ decision to voluntarily disclosure carbon information. Furthermore, â‘¢ has researched Chinese cases. 

→ However, I referred to â‘¡’s conclusion structure. 

(5)   Language : 

ETS should be presented in an extended version the first time it is used. It is suggested to proofreading the article to improve its readability and clarity. 

→ Page 2: ETS is presented in a full version. 

→ Proofreading: since I have been delaying in completing the revision, proofreading is still in process.  

Reviewer 2 Report

I believe that this study has identified an interesting and potentially quite important research topic. In this regard, I commend the authors. However, in its current form, I believe that the study is relatively simplistic in nature or immature, and would benefit from a strengthening of both the empirical and conceptual aspects. My specific comments appear below:

Conceptual:

1. I see no reason why the Korean setting should not be appropriate for the conduct of the study but please don't argue that the study advances the literature because prior studies have examined Western settings. Your arguments need to be established within the Korean regulatory setting (which my guess would lead to similar expectations are those in may settings with an ETS or mandatory emissions disclosure). In conjunction, please also provide more institutional background details around the Korean ETS

2. I believe that the authors are missing an opportunity when they develop their basic arguments and thereby hypotheses. An ETS is a setting where firm's are required to disclose their carbon emissions data in tonnes. This figure may in fact paint them in an unfavourable light; equally it may paint them in a favourable light. In this regard, I have to believe that the firm's disclosure response will relate to how the 'tonnes figure' portrays the firm's carbon-related performance. This I then believe moves the debate more towards what is said rather than how much is said.

3. In terms of foreign sales (extent of global operations), I believe that the 'where' is likely more important than just the how much. Different jurisdictions have different regulation around carbon emissions in place, and very different attitudes towards a firm's carbon footprint

4. I don't see media visibility as just a matter of the extent but also the general sentiment expressed in the media

5. Disappointingly, I don't see any mention of disclosure frameworks such as the TCFD that are starting form the basis of disclosure expectations in many countries.

Empirical:

The measures and model that the authors use is relatively basic. Here, I see a number of issues that could be addressed:

1. In terms of the measure of 'media visibility', the authors fail to consider the tone or sentiment of the underlying media coverage.  As an example Clarkson et al 2008 use the J-F measure that captures sentiment

2. I believe that the model is relatively incomplete - it certainly does not include measures around CapEx etc as are found in the literature

3. I do NOT see any mention of how the authors deal with outliers - for example, the minimum value of ROE is -1252% and the maximum value is 759% (these seem to be extreme values from my perspective)

I wish the authors well as they attempt to progress their study

Author Response

[Revision Summary Report]

Response to the Second Reviewer 

(1)  ETS’s institutional background 

→ Page 6~7: the Korean ETS’ institutional details are provided. 

(2)   Data in tons 

→ The amount of carbon emission was obtained from a governmental body “the Korean Greenhouse Gas Inventory and Research Center” that exclusively implements the Korean ETS. The subject companies of K ETS have a duty to report their emission on a yearly base. This means the data we used for the independent variable in this research is reliable and verified enough.

(3)   “where’ is more important. 

→ In the beginning, we considered obtaining foreign sales data by country but many Korean companies didn’t break down foreign sales by region. That’s why we were forced to gather total foreign sales amount on yearly basis.

(4)   Media visibility of general sentiment 

→ We believe that media sentiment you suggested would be a useful and effective variable. But given the purpose of our research, the extent of media exposure, either favorable or unfavorable, was more relevant to our sample companies.  

(5)   TCFD 

→ Although TCFD has been hotly debated in the field of carbon disclosure, most Korean companies primarily depend on sustainability reports for disclosing carbon-related information. The number of companies communicating their performance based on TCFD is very limited and also they include TCFD information in their sustainable report instead of releasing a separate report. 

That’s why our research has been intensively carried out with the contents analysis of sustainability reports based on a verified index. In this context, we decided not to highlight TCFD.

(6)   CapEx 

→ We used CapEx, as an additional control variable in our empirical test in accordance with your comment. You can see the robust result in Table 5,6 (Correlation, Regression result).  

(7)   Outliers 

→ As per outliers you suggested, we winsorized all variables except dummy variables at 1% and 99%. You can see the result in Table 4 Descriptive Statistic. THe results are essentially remain the same.

Reviewer 3 Report

General:

The comment concerns the statistical significance of variables in regression models, to which the Authors attach great importance. In empirical research, it is often the case that statistical significance is not that important. I believe that this is not defective. It is part of the current debate of scientists from various fields (excessive belief in p-value).

1. Abstract: Is abstract structured in accordance with the requirements of the Journal?

The topic of the paper is very interesting, however the paper has some limitations.  

The abstract is a bit confusing, and it is not in accordance with the title of the paper. The abstract provides a summary of the paper. The Authors could better highlight the contribution of the paper in the abstract.

The abstract should contain the paper’s objective, motivation, methodology, main findings, and contribution.

2. Motivation and contribution: Does the paper contain new and significant information that improve or build on existing research? What is theoretical, empirical and/or practical contribution of the paper?

The introduction is clear, however, it is worth highlighting (more clearly!) the research hypothesizes/questions and I did not find the main results obtained in the empirical analysis. I think that it is important to anticipate the main results in this part of the paper in order to allow the reader to better understand the analysis.

3. Literature: Does the paper present an adequate understanding of the relevant literature and refer to literature sources in the right way?

The lack of previous actual studies (in 2021, 2022). Please add and capture previous studies about this topic You need also highlight the finding from previous studies and try to tell the different approach of your study and then illustrate the contribution and novelty from your study.

For example,

Climate Risk with Particular Emphasis on the Relationship with Credit-Risk Assessment: What We Learn from Poland (2021), N Nehrebecka, Energies 14 (23), 8070

4. Methodology: Is the paper's idea built on an appropriate conceptual background? Is the research design and methods employed appropriate?

The research methodology is straightforward.

The Pooled OLS regression is not appropriate for the analyzed problem due to the – almost certain in that research field – fixed nature of the individual effects. Moreover in such a situation the additional endogeneity of the explanatory variables.

5. Findings: Are the results presented in a clear way, highlighting the novelty in the context of existing research?

Model diagnostics not performed.

6. Quality of Communication: Is the paper written in a clear way for readers and the used terminology is appropriate?

It is clearly written.

Author Response

[Revision Summary Report]

Response to the Third Reviewer 

(1)   Abstract 

→ Research purpose, methodology and contribution were added. 

(2)   Introduction 

→ Page 2: Research question was added. 

(3)   Literature 

→ I looked into the below paper. However, we found no particular relevance in the paper. Please understand that we did not included the paper in the reference.

  Climate Risk with Particular Emphasis on the Relationship with Credit-Risk Assessment (2021) 

(4)   Methodology 

â‘     Fixed nature of the individual effects : 

→ We conducted the Hausman test and the result proved significant fixed effect. That’s why we used year and industry dummy variables. We think this approach was relevant and valid. We described it in the first paragraph of empirical results section.

â‘¡   Endogeneity : 

→ We believe that in our research model, reverse causality between independent and dependent variables hardly happens. So we reassure you that our research model is unlikely to cause an endogeneity problem. Please understand our hard efforts to design the research model in a solid way.

(5)   Findings and Conclusions 

→ I made hard efforts to elaborate main findings in a clear way and highlighted the novelty and contribution in a firm way as well.  

Round 2

Reviewer 1 Report

I accept the paper in the current form.  Best of luck with your work

Reviewer 2 Report

Thank you for your responses - I have nothing further to add at this stage

Reviewer 3 Report

Dear Author,

Thank you very much. I accept.